# The Complete Plastome of ‘Mejhoul’ Date Palm: Genomic Markers and Varietal Identification

**DOI:** 10.3390/ijms262311603

**Published:** 2025-11-29

**Authors:** Monther T. Sadder, Anfal Alashoush, Nihad Alsmairat, Anwar Haddad

**Affiliations:** 1Department of Horticulture and Crop Science, School of Agriculture, University of Jordan, Amman 11942, Jordan; anfalalahoush@yahoo.com (A.A.); drnihad@ju.edu.jo (N.A.); 2Jordan Dates Association (JODA), 204 Zaytounah Complex A, Paris Street, Sweifieh, Amman 11185, Jordan; p@jodates.org

**Keywords:** date palm, ‘Mejhoul’, plastome, cultivar identification, fingerprinting, biodiversity

## Abstract

Next-generation sequencing technology was employed to read and assemble the complete plastid genome of the ‘Mejhoul’ date palm cultivar (*Phoenix dactylifera* L.). The genome consisted of 158,436 base pairs (bp) with a GC content of 37.24%, and it included 95 protein-coding genes, 44 tRNA genes, and eight rRNA genes. The plastome of five ‘Mejhoul’ genotypes from Jordan was compared with three genotypes from the USA, Morocco, and the UAE. It revealed 91 single-nucleotide polymorphisms (SNPs) and 23 insertions–deletions (InDels); the majority of them (62%) were located in intergenic regions, while the remaining variants were located in intragenic regions, including tRNA and rRNA genes. When the plastomes of all eight ‘Mejhoul’ genotypes were aligned, along with major cultivars ‘Barhee’ and ‘Khalas’, 24 SNPs and 23 InDels could be found. This would enable the development of a cultivar-specific fingerprint test for authentication. The phylogenetic tree was constructed using seventeen date palm cultivars. The phylogenetic analysis places ‘Mejhoul’ as a lineage derived within Clade I rather than as an early-diverging cultivar, suggesting it shares a more recent common ancestor with ‘Deglet Noor’ and ‘Barhee’.

## 1. Introduction

The date palm (*Phoenix dactylifera* L.) is one of 14 species under the genus *Phoenix* that are native to southern Europe, Africa, the Arabian Peninsula, and South Asia [1]. The date palm is a dioecious monocotyledonous fruit tree plant that can live for a long time and belongs to the Arecaceae (or Palmae) family [2,3]. Date palm has been domesticated for thousands of years in the centers of its origin, diversity, and domestication in the Middle East and North Africa [4].

It has historically been one of the most vital fruit crops in arid regions, serving as a primary source of revenue and a staple in the diet for local populations in many of the nations where it is grown, and has played key roles in those countries’ economies, societies, and environments [5,6]. It has been used as a food source, as well as in home building and landscaping [7].

Nowadays, date palm is considered one of the most important crops in many regions of the world with arid climates and high temperatures. It has taken an increasing commercial space in recent years worldwide to become a major food source in addition to its secondary products that are used in a wide range of industries [8]. Approximately 90% of date palms are cultivated in the Middle East and North Africa, with major producers including Egypt, Saudi Arabia, Algeria, Iran, Pakistan, Iraq, Sudan, UAE, Oman, Tunisia, Libya, Morocco, and Jordan [9]. In the past few centuries, dates have also been introduced into new production areas in Australia, the Indian Peninsula, Southern African countries, and South and North America [5]. Currently, there are nearly 100 million date palm trees worldwide, with around 62 million of them located in West Asia and North Africa [10]. This global distribution highlights the economic importance of date palms, as they serve not only as a staple food for local populations but also contribute significantly to the economies of the countries where they are grown [11].

The date palm king cultivar, ‘Mejhoul’, also known as ‘Medjool’ (Figure 1), is well known for large, elongated fruits with high sweetness and rich flavor, in addition to being a high-yielding cultivar that tolerates drought and moderate salinity. ‘Mejhoul’ originated from Morocco and was transferred to California, USA in 1927. From there, it was acquired after 45 years by local farms in the Middle East, mainly in the Jordan Valley [12]. In total, Jordan’s ‘Mejhoul’ production has reached fourth worldwide with more than 80% being produced for export; a growth attributed to the favorable climate conditions in the Jordan Valley, with warm winters, high summer heat, abundant sunshine, and enough irrigation water from the valley’s channel, creating an extended growing season with strong photosynthetic potential and reduced disease risk. These conditions are ideal for high-quality date production, especially for the ‘Mejhoul’, which are also supported with conducive investment opportunities in the date palm sector [13,14].

Date palm is being vegetatively propagated to maintain the genetic purity of the mother plant. Establishment of new farms relies mainly on two sources for new ‘Mejhoul‘ plantation: mother offshoots from female plants [15] and plant tissue culture-propagated plantlets [16]. Because the establishment of a date palm farm requires substantial investment and multiple years before fruiting, reliable authentication of the cultivar identity is essential. Traditionally, cultivar identification has relied on nuclear DNA markers, which are biparentally inherited and, therefore, represent both maternal and paternal alleles. Moreover, nuclear markers often show extensive diversity, distinguishing closely related cultivars in any dioecious plants, such as date palm, and may be affected by heterozygosity due to the different origins of the male genome and the inability to trace maternal lineages, which is crucial in date palm because only the female parent contributes offshoots and tissue-culture material.

In contrast, the plastome (chloroplast genome) is highly suitable for differentiating female lineages due to its uniparental (maternal) inheritance, conserved structure, and moderate evolutionary rate. Plastid genomes provide a stable framework for assessing variation among cultivars that share maternal genomic backgrounds. With the advent of next-generation sequencing (NGS), it is now possible to obtain complete plastome sequences with high depth and accuracy. Compared with classical markers, NGS-based plastome profiling provides maternally inherited variations, enabling the development of precise and robust diagnostic markers for cultivar authentication, especially for elite female clones like ‘Mejhoul’.

Moreover, as global demand for Mejhoul dates rises, additional challenges such as genetic uniformity, susceptibility to diseases and pests, and climate stress have emerged [17,18]. Addressing these issues requires advanced genetic research, particularly through tools like DNA fingerprinting, which can reveal genetic diversity and guide breeding programs [19]. Countries such as Saudi Arabia and Egypt have taken significant steps to incorporate genetic tools, enhancing disease resistance and sustainability in date palm cultivation [20]. In regions where Mejhoul date cultivation is extensive, such as the UAE, Saudi Arabia, and Morocco [21], genetic fingerprinting has helped to enhance the understanding of genetic variation within ‘Mejhoul’ populations, aiding in the development of superior cultivars [22]. The utilized DNA fingerprinting in date palm covers two major tools: simple sequence repeats (SSRs) [19,20], which are based on allelic variation in repeats among different genotypes. Although co-dominant in nature, it does not resolve, except for a limited number of polymorphic bands. On the major tool, SNPs are more powerful DNA markers, based usually on next-generation sequencing data [21], which delivers deeper genome coverage and high accuracy achieved with enough sequencing depth up to 1000×.

Research from countries like Morocco and Egypt has demonstrated the utility of DNA fingerprinting in identifying genetic diversity and aiding breeding programs for ‘Mejhoul’ dates [19]. For instance, studies in Morocco revealed specific genetic traits that have made ‘Mejhoul’ dates in that region more resilient to environmental stresses [23].

Therefore, this study aimed to sequence the entire ‘Mejhoul’ plastome, to identify the genetic diversity among current ‘Mejhoul’ plantations, and to establish a unique ‘Mejhoul’ fingerprint that is comparable with other major date cultivars.

## 2. Results

The major objective of this study was to generate a plastome sequence for a group of ‘Mejhoul’ individual accessions. It was possible to successfully assemble one circular for each of these accessions (five from Jordan, one from the USA, one from the UAE, and one from Morocco). The length of the complete plastome for ‘Mejhoul’ was 158,436 bp with 37.24% GC content (Figure 2). The plastid genome contained 147 genes: 95 protein-coding genes, 44 tRNA genes, and 8 rRNA genes. A total of 68 genes run on the outer circle (anticlockwise) and 79 run in the inner circle (clockwise) (Figure 2).

The generated plastomes were assumed to show similar genotypes. However, after multiple sequence alignment, many single-nucleotide polymorphisms (SNPs) and insertions–deletions (InDels) were detected (Table 1). In fact, 95 SNPs and 31 InDels were recorded along the entire plastome sequence. Most interestingly, genotypes for ‘Mejhoul’ 1, 2, 3, 4, and 5 had limited genetic changes, whereas ‘Mejhoul’ 7 and 8 were close to each other.

The plastome-based ML tree (Figure 3) supports the presence of genetic variation among the examined ‘Mejhoul’ genotypes; however, most internal relationships are modestly supported and, therefore, cannot be interpreted as a resolved branching order. ‘Mejhoul’ genotypes 1–5 form a broad group with generally low to moderate bootstrap values, indicating that their relative relationships remain unresolved. Within this group, only a few nodes (e.g., ‘Mejhoul’ genotypes 4 and 1) show moderate support, and no strict bifurcation pattern can be confidently inferred. In contrast, ‘Mejhoul’ genotypes 6, 7, and 8 form a well-supported cluster (100% bootstrap), although the branching within this cluster is only partially resolved; ‘Mejhoul’ genotypes 7 and 8 group together with moderate support (70%), while ‘Mejhoul’ genotype 6 clusters with the male genotype with strong support but without allowing for inference of evolutionary sequence. Overall, the topology highlights detectable plastome differences among accessions but does not support a definitive order of divergence.

On the other hand, a plastome multiple sequence alignment was generated between all ‘Mejhoul’ genotypes 1–8 and two important date palm cultivars, ‘Barhee’ from the United Arab Emirates (CM018784) and ‘Khalas’ from Saudi Arabia (NC_013991) (Table 2). The data was mined for SNPs and InDels, which would show 100% consensus among all eight ‘Mejhoul’ genotypes, while they are different in date palm cultivars ‘Barhee’ and ‘Khalas’, taking into consideration that they are identical in both cultivars ‘Barhee’ and ‘Khalas’. In total, there were 47 DNA changes (24 SNPs and 23 InDel).

Another phylogenetic tree was constructed using the plastome of ‘Mejhoul’ 2 and the available plastomes in GenBank (NCBI, 2025), including date palm cultivars ‘Barhee’ from the United Arab Emirates (CM018784), ‘Khalas’ from Saudi Arabia (NC_013991), ‘DP00001’ from Pakistan (FJ212316), and ‘SUPG02’ from India (MF176947) (Figure 4). In this tree, ‘SUPG02’ from India did not cluster closely with any of the other cultivars, while the remaining cultivars formed a major cluster. Within this cluster, the final subclade grouped both ‘Mejhoul’ 2 and ‘Barhee’ with moderate bootstrap support (85%). However, the other major subclade grouped ‘Khalas’ and ‘DP00001’ (96%), indicating a closer relationship between these two genotypes.

Although Genbank (NCBI, 2025) had a limited number of date palm plastome sequences, there were plenty of whole genome sequences (WGSs) deposited as short read archives (SRA). Data for more than 100 well-known date palm cultivars from around the world are available at Genbank. The top seventeen cultivars were selected to run a comprehensive phylogenetic tree (Figure 5). The Maximum Likelihood tree resolved into three major groups. Group I exhibited stronger statistical support, with subgroups including ‘Deglet Noor’, ‘Mejhoul’ 2, and ‘Barhee’ (91% bootstrap), though some portions, particularly the relationships involving ‘Zahdi’ and ‘Nebtet Ali’, lacked sufficient support. Group II, which contained cultivars such as ‘Chichi’, ‘Brem’, ‘Khenezi’, ‘Mabrouma’, ‘Soukari’, ‘Sagai’, and ‘Khadrawy’, did not form a well-supported clade, and the support for these cultivars being grouped together was weak, suggesting a more nuanced relationship between them. Finally, Group III included ‘Safri’, ‘Sultana’, ‘Zagloul’, and ‘Ajwa’; this grouping also lacked strong bootstrap support, and their placement should be reconsidered as part of a larger phylogenetic analysis rather than being treated as a distinct clade.

A pairwise comparison analysis was listed in an array (Figure 6). The most similar cultivar for ‘Mejhoul’ is ‘Deglet Noor’ with only 65 bp differences (the lowest number among all other cultivars) and only 16 gaps. On the contrary, ‘Berhee’ displayed the most abundant differences with all date palm cultivars (17,783–17,992) and also gaps (11,028–11,185). This was followed by ‘Sultana’ with moderate bp differences (194–360) and gaps (119–212), as compared with other date palm cultivars except for the most different cultivar, ‘Berhee’.

## 3. Discussion

The present study revealed for the first time the genetically diverse genotypes of the ‘Mejhoul’ date palm from the available stands in the region (Table 1). This was not expected, as the history of the ‘Mejhoul’ expansion was thought to be clonal the entire time. Consequently, this would imply multiple origins of the current date palm plantations. In a comprehensive study to assess ‘Mejhoul’ genetic diversity in its original site in Morocco [24], 66 different accessions from Morocco and 4 from California, USA, were investigated using amplified fragment length polymorphism (AFLP). The Moroccan accession showed 21% genetic diversity, indicating that it represents a landrace rather than a uniform cultivar. Consistent with this, our plastome data also showed unexpected variation among the five investigated ‘Mejhoul’ genotypes, while the US accession was placed separately in our plastome tree (Figure 3). Likewise, the four accessions from the USA were also resolved separately from the Moroccan accession [24]. If the origin (Morocco) of ‘Mejhoul’ showed intra-varietal variation within a landrace, what would the second station (USA) of the ‘Mejhoul’ intercontinental journey show [12]? In fact, a similar study was conducted for 23 US accessions of ‘Mejhoul’, also using AFLP DNA markers [25]. Although the main propagation means were offshoots in the US, which is similar to the Moroccan study [24] and our results (Figure 3), huge genetic diversity was revealed among the US investigated accessions. The authors argued the possibility of high-mutation build-up in different regions throughout the long history of cultivation [25]. This genetic variation was reflected again when the ‘Mejhoul’ journey continued from the USA to the Middle East (Figure 3).

The major issue with whole-genome DNA markers like AFLP is that they cover both homologous chromosome pairs from the ancestry parents, both combined to form the current tree. Why this is a major issue for date palm is because it is a dioecious plant, where female plants are the limiting factor for determining the cultivar characteristics, irrespective of the pollenating male plants, and also irrespective of the original male parent of the current female tree. Therefore, restricting the analysis to the female parent, e.g., organelle genomes, similar to this study, would scale down the genetic noise spiked by multiple origins of male parents. In fact, studies have indicated that dioecious clades might diversify more than expected [26]. Moreover, dioecious plants can display a sort of pollination selection (acting pre- or post-zygotically), which may modify the outcome of pollination [27]. In this regard, it is worth mentioning that it is possible to induce hermaphrodism in vitro for date palm female flowers, suggesting that dioecious plants come from a hermaphrodite ancestor [28]. Moreover, cases of natural date palm hermaphroditism (male and female inflorescences on the same tree), although very rare, have been described [29,30]. On the other hand, gender specialization appears to be favored in resource-poor environments, regardless of which pathway is taken to dioecy [31]. Therefore, it is expected to detect a swinging state between dioecy (extended period) and monoecy (very short period) in any date palm grove over time. This plausible scenario would hinder the interpreting of date palm genetic diversity. However, it would explain plastome heteroplasmy [32].

Several other DNA fingerprinting methods were investigated to assess genetic variability and cultivar identification in date palms. The genetic variability of four date palm cultivars in Saudi Arabia (Mejhoul b1, Sugayi b1, Khalas b1, Sukkari b1) was assessed using Random Amplified Polymorphic DNA (RAPD) and Inter-Simple Sequence Repeat (ISSR) markers [33]. The UPGMA cluster analysis identified two clusters: Sukkari b1 (0.55 similarity) in cluster A, and Mejhoul b1, Sugayi b1, and Khalas b1 (0.66–0.85 similarity) in cluster B [33], whereas our plastome data indicated that ‘Mejhoul’ 2 and ‘Khalas’ do not appear closely related in our tree (Figure 4). Similar studies employed whole-genome DNA markers, including the use of AFLP markers in the genetic analysis of Egyptian date palm cultivars [34], ISSR markers in the genetic analysis of Moroccan date palm cultivars [35], and SSR markers in the genetic analysis of Ethiopian date palm cultivars [20]. Unfortunately, all these whole genome markers would be highly influenced by their nature, where the genetic diversity is spiked with multiple and diverse-ancestry male parents.

In the present study, plastome comparisons revealed a relatively close maternal relationship between ‘Mejhoul’ and ‘Barhee’ (Figure 4). This finding is relevant because young offshoots or tissue-culture plantlets of these cultivars are often difficult to distinguish morphologically during the establishment of new plantations. Nonetheless, several unique SNPs and InDels were identified in our plastome dataset that can assist in the reliable varietal identification (Table 2). Prior SNP-based work has reported variable relationships involving ‘Mejhoul’, including closer similarity with ‘Deglet Noor’ [36], whereas other studies found these two cultivars to be distantly related [21]. Such differences may reflect the variation in sampled accessions or differences in SNP platforms. In our comprehensive plastome tree (Figure 5), the relative proximity among cultivars should be interpreted cautiously, as several internal nodes lacked strong statistical support. While ‘Deglet Noor’ appeared near ‘Mejhoul’ in our plastome-based analysis, the low support for several branches indicates that these relationships should not be viewed as definitive and require further sampling and higher-resolution markers for confirmation.

## 4. Materials and Methods

### 4.1. Plant Material

Five accessions of ‘Mejhoul’ date palms were selected from the oldest palm orchards located in the Jordan Valley. Leaf samples were acquired for each accession (1,2, 3, 4, and 5), which were from different farms. They were established forty years ago and were sourced at that time from the United States (California).

Total genomic DNA was isolated with the Genomic DNA Preparation Kit (BioFact, Daejeon, Republic of Korea) according to the manufacturer’s instructions. Leaf samples were ground using metal beads with the BeadBlaster 24 Microtube Homogenizer (Benchmark, Tempe, AZ, USA). DNA was checked using 0.8% agarose gel electrophoresis with TAE running buffer.

### 4.2. NGS and Assembly

Genomic DNA was sequenced using NovaSeq 6000 and 151 bp paired-end reads (Illumina, San Diego, CA, USA). The reads were paired and mapped to the reference date palm plastome ‘Khalas’ from Saudi Arabia (NC_013991) [37]. CLC Genomics Workbench (Redwood City, CA, USA) was used for the assembly of the ‘Mejhoul’ plastomes. Mapping parameters were: No masking, Match score = 1, Mismatch cost = 2, Insertion cost = 3, Deletion cost = 3, Length fraction = 0.5, Similarity fraction = 0.8, Auto-detect paired distances. For consensus parameters: Remove regions with low coverage, Join after removal, Transfer annotations from reference. The ‘Mejhoul’ date palm plastome was visualized using OGDRAW [38]. The ‘Mejhoul’ 1–5 sequence was deposited in Genbank under accession numbers PX560697, PX574330-PX574333, respectively [39]. The coverage depth indicated enough coverage for ‘Mejhoul’ accessions 1–8 (Figure 7). For SNP and InDel detection, multiple alignments were generated, followed by the mining of DNA changes in each alignment group.

### 4.3. Phylogenetic Analysis

Additional plastomes for three additional genotypes of ‘Mejhoul’ were retrieved from short read archives (SRA) deposited at Genbank [39] for date palm whole genome sequencing studies: ‘Mejhoul’ 6 from USA (SRR121603) [21], ‘Mejhoul’—Male from USA (SRR121598-SRR121601) [21], ‘Mejhoul’ 7 from UAE (SSR10121113) [40], and ‘Mejhoul’ 8 from Morocco (SRR2559387) [22]. All eight ‘Mejhoul’ sequences were aligned using CLC Genomics Workbench (Redwood City, CA, USA), multiple alignment parameters were: Gap open cost = 10.0, Gap extension cost = 1.0, End gap cost = As any other. Followed by bootstrapping 1000 times, and employed to generate a maximum-likelihood phylogenetic tree using PHYLIP [41], tree construction parameters were: starting tree method = Neighbor Joining, Nucleotide substitution model = Felsenstein 81 [42], Number of substitution rate categories = 4, Gamma distribution parameter = 1.0, Estimate substitution rate parameter(s) = Yes, Estimate topology = Yes, Estimate gamma distribution parameter = No. Moreover, another maximum-likelihood phylogenetic tree was generated with the additional date palm plastomes: ‘Barhee’ from United Arab Emirates (CM018784), ‘DP00001’ from Pakistan (FJ212316), ‘Khalas’ from Saudi Arabia (NC_013991), and ‘SUPG02’ from India (MF176947). In a comprehensive phylogenetic analysis, SRAs covering the top 15 date palm cultivars in terms of worldwide production were retrieved from Genbank (NBCI, 2025). The selected cultivars, along with their SRA numbers, are: Deglet Noor (SRR2577995), Ajwa (SRR10121077), Nebtet Ali (SRR10121101), Sagai (SRR10121107), Safri (SRR10121132), Soukari (SRR10121134), Rothana (SRR10121136), Zagloul (SRR10121157), Zahdi (SRR10121162), Sultana (SRR10121164), Mabrouma (SRR10121165), Brem (SRR10121166), Chichi (SRR10121194), Khadrawy (SRR10121207), and Khenezi (SSR10121187). Sequence reads were mapped to ‘Khalas’ from Saudi Arabia (NC_013991) [37].

## 5. Conclusions

This study revealed detectable plastome heterogeneity among the examined ‘Mejhoul’ genotypes. The variation reflects differences among maternal lineages rather than pollen-derived nuclear diversity, underscoring the value of plastid genomes for tracing female ancestry in clonal cultivars. The results indicate that historical dissemination of the ‘Mejhoul’ across regions has preserved multiple maternal lines, leading to subtle but consistent plastome divergence. These findings demonstrate that plastome-based NGS profiling provides a robust framework for distinguishing female lineages, assessing clonal identity, and improving cultivar authentication in date palm.

## Figures and Tables

**Figure 1 ijms-26-11603-f001:**
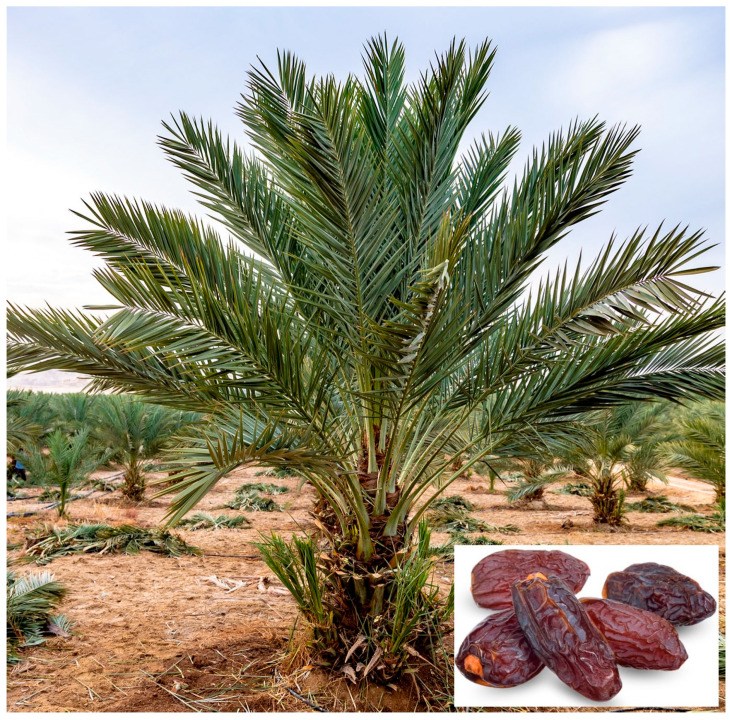
A representative tree photo for ‘Mejhoul’ date palm cultivar. The mature fruits are shown in the bottom right. Mejhoul Village, Al-Karamah, Jordan Valley, Jordan.

**Figure 2 ijms-26-11603-f002:**
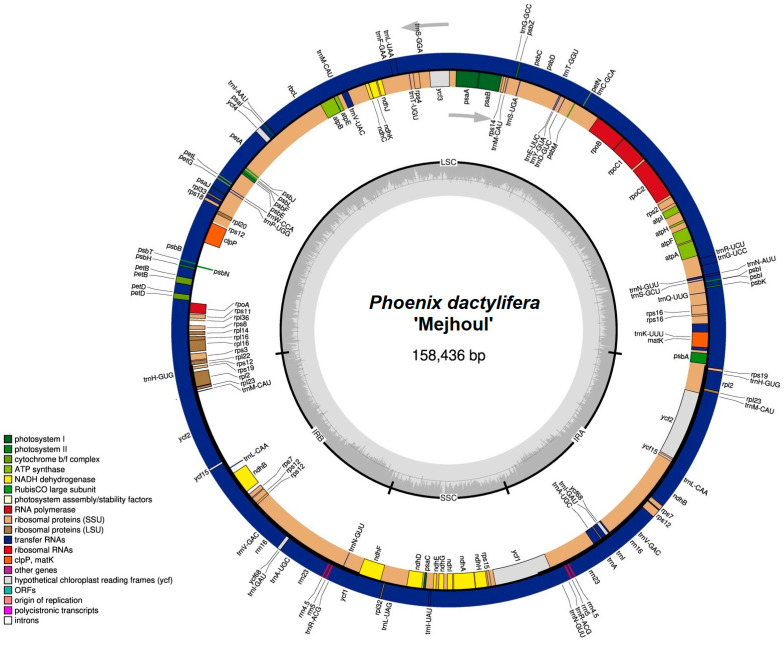
The complete plastome structure of ‘Mejhoul’ accession 2 from Jordan. Color key represents major genes and gene families. The inner gray rings represents GC content.

**Figure 3 ijms-26-11603-f003:**
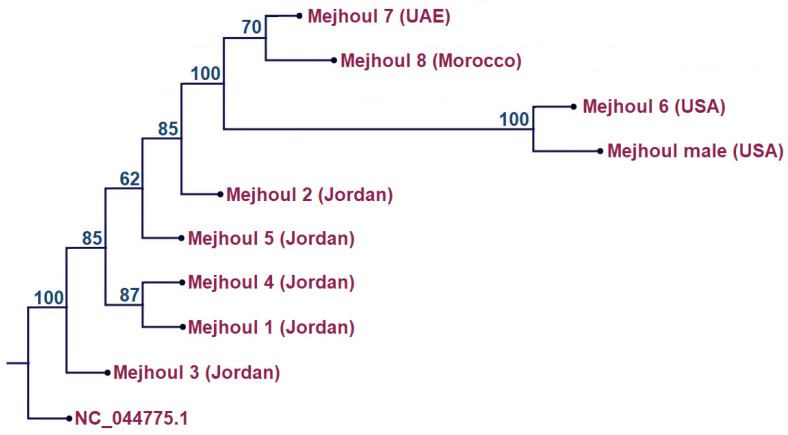
Maximum likelihood phylogenetic tree of eight ‘Mejhoul’ genotypes. ‘Mejhoul’ 1–5 from Jordan (PX560697, PX574330-PX574333), ‘Mejhoul’ 7 from UAE (SSR10121113), ‘Mejhoul’ 8 from Morocco (SRR2559387), ‘Mejhoul’ male from USA (SRR121603), and ‘Mejhoul’ male from USA (SRR121598). *Zingiber officinale* (NC_044775.1) was used as outgroup. Percentage bootstrap values are given on each branch (1000 runs).

**Figure 4 ijms-26-11603-f004:**
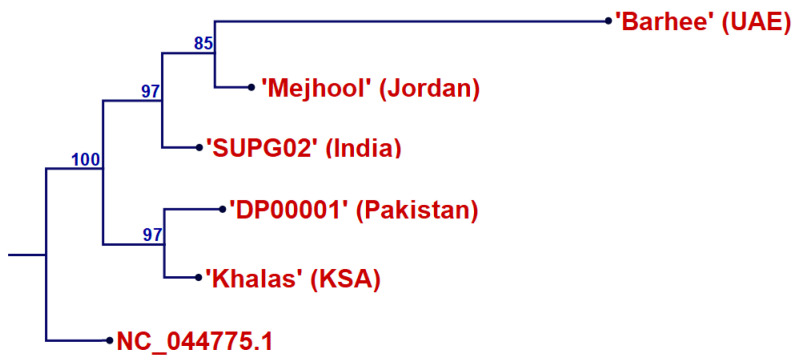
Maximum likelihood phylogenetic tree of ‘Mejhoul’ 2 (PX574330) from Jordan, along with ‘Barhee’ from United Arab Emirates (CM018784), ‘DP00001’ from Pakistan (FJ212316), ‘Khalas’ from Saudi Arabia (NC_013991), and ‘SUPG02’ from India (MF176947). *Zingiber officinale* (NC_044775.1) was used as outgroup. Percentage bootstrap values are given on each branch (1000 runs).

**Figure 5 ijms-26-11603-f005:**
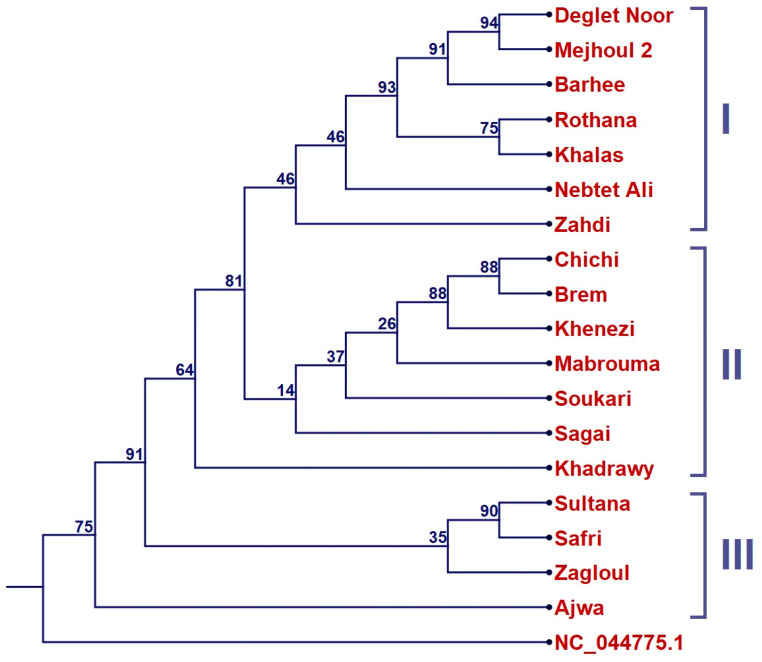
Maximum likelihood phylogenetic tree of ‘Mejhoul’ 2 from Jordan along with ‘Barhee’ from United Arab Emirates (CM018784), ‘Khalas’ from Saudi Arabia (NC_013991), and assembled plastomes for date palm cultivars (including SRA number): Deglet Noor (SRR2577995), Ajwa (SRR10121077), Nebtet Ali (SRR10121101), Sagai (SRR10121107), Safri (SRR10121132), Soukari (SRR10121134), Rothana (SRR10121136), Zagloul (SRR10121157), Zahdi (SRR10121162), Sultana (SRR10121164), Mabrouma (SRR10121165), Brem (SRR10121166), Chichi (SRR10121194), Khadrawy (SRR10121207), and Khenezi (SSR10121187). *Zingiber officinale* (NC_044775.1) was used as outgroup. Three major groups were resolved; I, II and III. Percentage bootstrap values are given on each branch (1000 runs).

**Figure 6 ijms-26-11603-f006:**
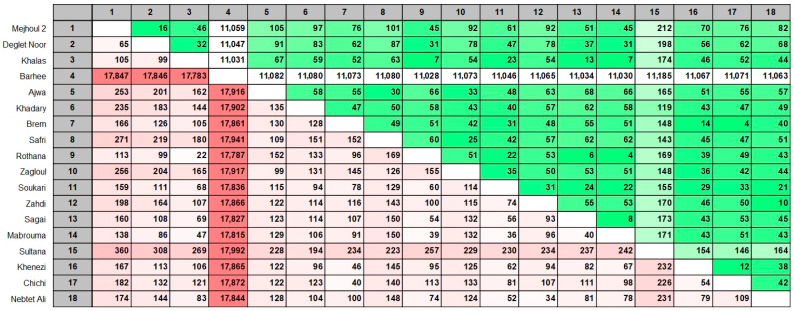
Pairwise comparison between all investigated date palm cultivars. Upper comparison in green represents gaps. Lower comparison in red represents base pair differences.

**Figure 7 ijms-26-11603-f007:**
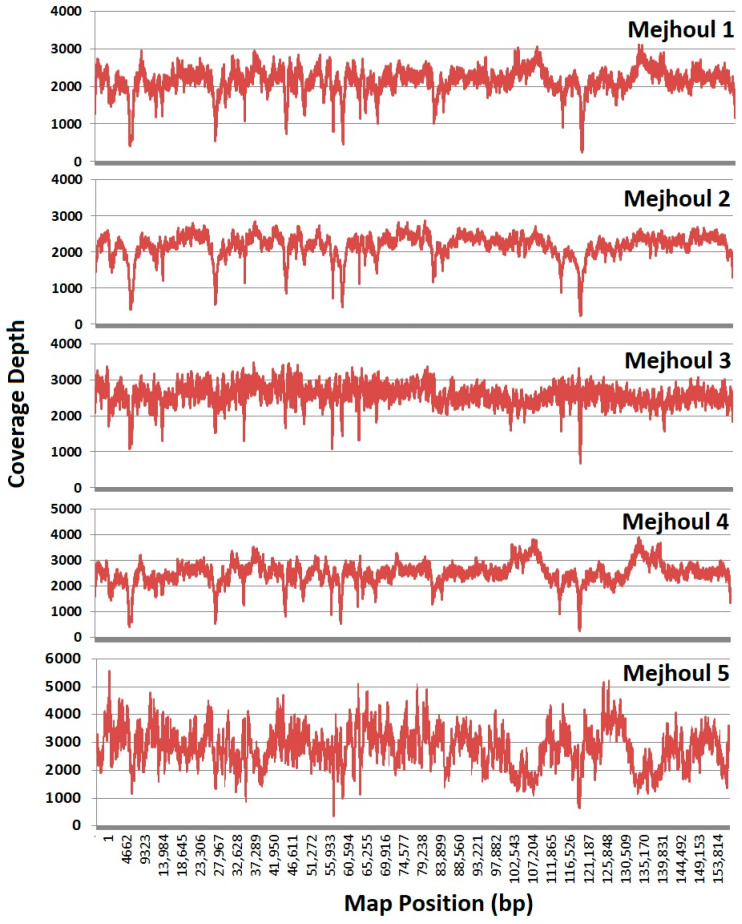
Sequencing depth and coverage map for ‘Mejhoul’ plastomes 1–5.

**Table 1 ijms-26-11603-t001:** Detected SNPs and InDels between the aligned plastome sequences of ‘Mejhoul’ accessions 1, 2, 3, 4, 5, 6, 7, and 8, along with map position (bp) and sequence nature. Key is shown in the bottom.

1	2	3	4	5	6	7	8	Map	Sequence	1	2	3	4	5	6	7	8	Map	Sequence	1	2	3	4	5	6	7	8	Map	Sequence
**A**	**A**	**A**	**A**	**A**	**A**	**A**	**T**	3916	tRNA-UUU	**-**	**-**	**A**	**-**	**-**	**-**	**-**	**-**	61,398	tRNA-AAU	**A**	**A**	**A**	**A**	**A**	**T**	**A**	**A**	84,439	RPL16
**A**	**A**	**A**	**A**	**A**	**A**	**A**	**T**	3921	tRNA-UUU	**T**	**T**	**T**	**T**	**T**	**C**	**C**	**C**	65,544	Intergenic	**A**	**A**	**A**	**A**	**A**	**T**	**A**	**A**	84,442	RPL16
**-**	**-**	**-**	**-**	**-**	**-**	**-**	**G**	3925	tRNA-UUU	**A**	**A**	**A**	**A**	**A**	**C**	**C**	**C**	65,546	Intergenic	**G**	**G**	**G**	**G**	**G**	**T**	**G**	**G**	104,863	rRNA16
**A**	**A**	**A**	**A**	**A**	**A**	**A**	**T**	3939	tRNA-UUU	**G**	**G**	**G**	**G**	**G**	**T**	**T**	**T**	65,549	Intergenic	**A**	**A**	**A**	**A**	**A**	**C**	**A**	**A**	104,867	rRNA16
**-**	**-**	**-**	**-**	**-**	**-**	**-**	**T**	3953	tRNA-UUU	**C**	**C**	**C**	**C**	**C**	**T**	**T**	**T**	65,550	Intergenic	**T**	**T**	**T**	**T**	**T**	**A**	**T**	**T**	104,896	rRNA16
**A**	**A**	**A**	**A**	**A**	**T**	**A**	**A**	4841	Intergenic	**T**	**T**	**T**	**T**	**T**	**G**	**G**	**G**	65,551	Intergenic	**T**	**T**	**T**	**T**	**T**	**C**	**T**	**T**	104,873	rRNA16
**A**	**A**	**A**	**A**	**A**	**T**	**A**	**A**	4845	Intergenic	**C**	**C**	**C**	**C**	**C**	**T**	**T**	**T**	65,552	Intergenic	**C**	**C**	**C**	**C**	**C**	**T**	**C**	**C**	104,876	rRNA16
**A**	**A**	**A**	**A**	**A**	**T**	**A**	**A**	4848	Intergenic	**A**	**A**	**A**	**A**	**A**	**T**	**T**	**T**	65,554	Intergenic	**G**	**G**	**G**	**G**	**G**	**C**	**G**	**G**	104,878	rRNA16
**A**	**A**	**A**	**A**	**A**	**T**	**A**	**A**	4849	Intergenic	**A**	**A**	**A**	**A**	**A**	**G**	**G**	**G**	65,555	Intergenic	**G**	**G**	**G**	**G**	**G**	**C**	**G**	**G**	104,879	rRNA16
**G**	**G**	**G**	**G**	**G**	**T**	**G**	**G**	4856	Intergenic	**T**	**T**	**T**	**T**	**T**	**A**	**A**	**A**	65,556	Intergenic	**C**	**C**	**C**	**C**	**C**	**T**	**C**	**C**	104,881	rRNA16
**T**	**T**	**T**	**T**	**T**	**-**	**-**	**-**	4870	Intergenic	**C**	**C**	**C**	**C**	**C**	**T**	**T**	**T**	65,557	Intergenic	**G**	**G**	**G**	**G**	**G**	**A**	**G**	**G**	120,387	Intergenic
**T**	**T**	**T**	**T**	**T**	**-**	**-**	**-**	4871	Intergenic	**A**	**A**	**A**	**A**	**A**	**T**	**T**	**T**	65,558	Intergenic	**A**	**A**	**A**	**A**	**A**	**T**	**A**	**A**	120,391	Intergenic
**T**	**-**	**T**	**-**	**T**	**-**	**-**	**-**	4872	Intergenic	**A**	**A**	**A**	**A**	**A**	**G**	**G**	**G**	65,560	Intergenic	**T**	**T**	**T**	**T**	**T**	**A**	**T**	**T**	120,392	Intergenic
**-**	**-**	**-**	**-**	**-**	**A**	**A**	**A**	9220	Intergenic	**C**	**C**	**C**	**C**	**C**	**A**	**A**	**A**	65,561	Intergenic	**T**	**T**	**T**	**T**	**T**	**A**	**T**	**T**	120,393	Intergenic
**-**	**-**	**-**	**-**	**-**	**T**	**T**	**T**	9221	Intergenic	**A**	**A**	**A**	**A**	**A**	**G**	**G**	**G**	65,562	Intergenic	**A**	**A**	**A**	**A**	**A**	**G**	**A**	**A**	120,395	Intergenic
**G**	**G**	**G**	**G**	**G**	**A**	**G**	**G**	9225	Intergenic	**A**	**A**	**A**	**A**	**A**	**C**	**C**	**C**	65,563	Intergenic	**T**	**T**	**T**	**T**	**T**	**C**	**T**	**T**	120,397	Intergenic
**C**	**C**	**C**	**C**	**C**	**T**	**C**	**C**	9228	Intergenic	**G**	**G**	**G**	**G**	**G**	**T**	**T**	**T**	65,566	Intergenic	**C**	**C**	**C**	**C**	**C**	**A**	**C**	**C**	120,398	Intergenic
**C**	**C**	**C**	**C**	**C**	**A**	**A**	**C**	9251	Intergenic	**G**	**G**	**G**	**G**	**G**	**A**	**A**	**A**	65,568	Intergenic	**A**	**A**	**A**	**A**	**A**	**G**	**A**	**A**	120,400	Intergenic
**T**	**T**	**T**	**T**	**T**	**A**	**A**	**T**	9254	Intergenic	**T**	**T**	**T**	**T**	**T**	**T**	**T**	**-**	69,867	Intergenic	**T**	**T**	**T**	**T**	**T**	**-**	**T**	**T**	120,404	Intergenic
**T**	**T**	**T**	**T**	**T**	**A**	**A**	**T**	9261	Intergenic	**C**	**C**	**C**	**C**	**C**	**C**	**C**	**T**	69,870	Intergenic	**A**	**A**	**A**	**A**	**A**	**T**	**A**	**A**	120,409	Intergenic
**T**	**T**	**T**	**T**	**T**	**A**	**A**	**T**	9263	Intergenic	**T**	**T**	**T**	**T**	**T**	**T**	**T**	**G**	69,871	Intergenic	**G**	**G**	**G**	**G**	**G**	**T**	**A**	**T**	120,424	Intergenic
**T**	**T**	**T**	**T**	**T**	**A**	**A**	**T**	9264	Intergenic	**A**	**A**	**A**	**A**	**A**	**A**	**A**	**T**	69,874	Intergenic	**A**	**A**	**A**	**A**	**A**	**A**	**-**	**-**	120,453	tRNA-UAU
**T**	**T**	**T**	**T**	**T**	**A**	**A**	**T**	9269	Intergenic	**C**	**C**	**C**	**C**	**C**	**C**	**C**	**T**	69,875	Intergenic	**-**	**-**	**-**	**-**	**-**	**-**	**T**	**-**	120,719	Intergenic
**T**	**T**	**T**	**T**	**T**	**A**	**A**	**A**	9286	Intergenic	**-**	**-**	**-**	**-**	**-**	**-**	**-**	**G**	69,878	Intergenic	**G**	**G**	**G**	**G**	**G**	**T**	**G**	**G**	130,157	Ycf1
**T**	**T**	**T**	**T**	**T**	**A**	**T**	**T**	9288	Intergenic	**C**	**C**	**C**	**C**	**C**	**C**	**C**	**T**	69,890	Intergenic	**A**	**A**	**A**	**A**	**A**	**A**	**T**	**T**	134,605	rRNA23
**C**	**C**	**C**	**C**	**C**	**C**	**A**	**C**	9289	Intergenic	**T**	**T**	**T**	**T**	**T**	**T**	**A**	**T**	83,890	Intergenic	**G**	**G**	**G**	**G**	**G**	**G**	**C**	**C**	134,608	rRNA23
**A**	**A**	**A**	**A**	**A**	**C**	**A**	**A**	47,238	Intergenic	**A**	**A**	**-**	**A**	**A**	**C**	**C**	**C**	84,413	RPL16	**C**	**C**	**C**	**C**	**C**	**C**	**T**	**T**	134,609	rRNA23
**-**	**-**	**-**	**-**	**-**	**C**	**-**	**-**	47,257	Intergenic	**T**	**T**	**-**	**T**	**T**	**T**	**T**	**T**	84,414	RPL16	**T**	**T**	**T**	**T**	**T**	**T**	**A**	**A**	134,610	rRNA23
**T**	**T**	**T**	**T**	**T**	**C**	**T**	**T**	47,259	Intergenic	**T**	**T**	**-**	**T**	**T**	**T**	**T**	**T**	84,415	RPL16	**A**	**A**	**A**	**A**	**A**	**A**	**G**	**G**	134,613	rRNA23
**G**	**G**	**G**	**G**	**G**	**A**	**G**	**G**	47,260	Intergenic	**T**	**T**	**-**	**T**	**T**	**T**	**T**	**T**	84,416	RPL16	**A**	**A**	**A**	**A**	**A**	**A**	**G**	**G**	134,614	rRNA23
**A**	**A**	**A**	**A**	**A**	**C**	**A**	**A**	47,281	Intergenic	**T**	**T**	**-**	**T**	**T**	**T**	**T**	**A**	84,417	RPL16	**C**	**C**	**C**	**C**	**C**	**C**	**T**	**T**	134,619	rRNA23
**C**	**C**	**C**	**C**	**C**	**T**	**C**	**C**	47,367	Intergenic	**T**	**T**	**-**	**T**	**T**	**T**	**T**	**T**	84,418	RPL16	**G**	**G**	**G**	**G**	**G**	**G**	**A**	**A**	134,625	rRNA23
**A**	**A**	**A**	**A**	**A**	**A**	**T**	**A**	47,398	Intergenic	**T**	**T**	**-**	**T**	**T**	**T**	**T**	**A**	84,419	RPL16	**C**	**C**	**C**	**C**	**C**	**C**	**T**	**T**	134,626	rRNA23
**-**	**-**	**-**	**-**	**-**	**A**	**A**	**A**	47,410	Intergenic	**T**	**T**	**A**	**T**	**T**	**T**	**T**	**T**	84,420	RPL16	**C**	**C**	**C**	**C**	**C**	**C**	**A**	**A**	134,632	rRNA23
**-**	**-**	**-**	**-**	**-**	**T**	**T**	**T**	47,411	Intergenic	**-**	**-**	**T**	**-**	**-**	**T**	**T**	**T**	84,421	RPL16	**G**	**G**	**G**	**G**	**G**	**A**	**G**	**G**	139,767	rRNA16
**A**	**A**	**A**	**A**	**A**	**T**	**A**	**A**	47,419	Intergenic	**-**	**-**	**T**	**-**	**-**	**T**	**T**	**T**	84,422	RPL16	**C**	**C**	**C**	**C**	**C**	**G**	**C**	**C**	139,769	rRNA16
**C**	**C**	**C**	**C**	**A**	**A**	**C**	**C**	58,969	Intergenic	**-**	**-**	**T**	**-**	**-**	**T**	**T**	**T**	84,423	RPL16	**C**	**C**	**C**	**C**	**C**	**G**	**C**	**C**	139,770	rRNA16
**C**	**C**	**C**	**C**	**-**	**C**	**A**	**C**	58,970	Intergenic	**-**	**-**	**T**	**-**	**-**	**T**	**T**	**T**	84,424	RPL16	**G**	**G**	**G**	**G**	**G**	**A**	**G**	**G**	139,772	rRNA16
**-**	**-**	**-**	**-**	**-**	**A**	**-**	**-**	58,980	Intergenic	**-**	**-**	**T**	**-**	**-**	**T**	**T**	**T**	84,425	RPL16	**A**	**A**	**A**	**A**	**A**	**G**	**A**	**A**	139,775	rRNA16
**C**	**T**	**C**	**C**	**T**	**T**	**T**	**T**	61,317	tRNA-AAU	**-**	**-**	**T**	**-**	**-**	**T**	**T**	**T**	84,426	RPL16	**A**	**A**	**A**	**A**	**A**	**T**	**A**	**A**	139,779	rRNA16
**A**	**T**	**A**	**A**	**A**	**T**	**T**	**T**	61,386	tRNA-AAU	**-**	**-**	**T**	**-**	**-**	**T**	**T**	**T**	84,427	RPL16	**T**	**T**	**T**	**T**	**T**	**G**	**T**	**T**	139,781	rRNA16
**A**	**A**	**T**	**A**	**A**	**A**	**A**	**A**	61,391	tRNA-AAU	**A**	**A**	**A**	**A**	**A**	**T**	**A**	**A**	84,437	RPL16	**C**	**C**	**C**	**C**	**C**	**A**	**C**	**C**	139,785	rRNA16

KEY: Intergenic, rRNA16, tRNA, RPL.

**Table 2 ijms-26-11603-t002:** Unique differentiating SNPs and InDels between the aligned plastome sequences of ‘Mejhoul’ (all accessions 1, 2, 3, 4, 5, 6, 7, and 8) compared with ‘Barhee’ and ‘Khalas’, along with map position (bp).

‘Mejhoul’ 1–8	‘Barhee’ & ‘Khalas’	Map	‘Mejhoul’ 1–8	‘Barhee’ & ‘Khalas’	Map	‘Mejhoul’ 1–8	‘Barhee’ & ‘Khalas’	Map
** **C** **	** **T** **	17	** **-** **	** **A** **	14,785	** **-** **	** **A** **	64,985
** **T** **	** **A** **	112	** **-** **	** **A** **	14,786	** **C** **	** **G** **	66,944
** **A** **	** **G** **	152	** **-** **	** **T** **	14,787	** **T** **	** **C** **	67,889
** **G** **	** **A** **	1433	** **T** **	** **C** **	20,212	** **T** **	** **G** **	69,478
** **T** **	** **-** **	3842	** **C** **	** **T** **	22,579	** **C** **	** **A** **	70,574
** **T** **	** **C** **	4741	** **T** **	** **C** **	24,138	** **T** **	** **-** **	72,593
** **A** **	** **G** **	5038	** **-** **	** **A** **	29,890	** **T** **	** **-** **	72,857
** **A** **	** **G** **	7164	** **-** **	** **T** **	31,439	** **G** **	** **A** **	73,745
** **T** **	** **-** **	7451	** **-** **	** **A** **	33,299	** **G** **	** **A** **	75,080
** **A** **	** **G** **	8279	** **C** **	** **A** **	35,903	** **T** **	** **-** **	77,603
** **G** **	** **T** **	9094	** **C** **	** **T** **	36,168	** **G** **	** **T** **	83,409
** **-** **	** **T** **	13,895	** **A** **	** **-** **	60,920	** **A** **	** **C** **	83,410
** **C** **	** **A** **	14,621	** **-** **	** **A** **	64,981	** **T** **	** **C** **	84,422
** **-** **	** **T** **	14,645	** **-** **	** **A** **	64,982	** **-** **	** **T** **	84,831
** **-** **	** **A** **	14,783	** **-** **	** **T** **	64,983	** **-** **	** **T** **	86,227
** **-** **	** **T** **	14,784	** **-** **	** **G** **	64,984			

## Data Availability

The original contributions presented in this study are included in the article material. Further inquiries can be directed to the corresponding author.

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
