# Peer review of "The Complete Plastome of ‘Mejhoul’ Date Palm: Genomic Markers and Varietal Identification"

_ijms, 2025, doi:10.3390/ijms262311603_

Round 1

Reviewer 1 Report

Comments and Suggestions for Authors

Dear Author:

After reviewing the manuscript, the evaluation comments are as follows. In this study, next-generation sequencing (NGS) technology was employed to complete the plastid genome assembly of the well-known date palm variety 'Mejhoul', accurately presenting key foundational data such as genome size, GC content, and gene count. Comparative analyses of the plastid genome were conducted between 'Mejhoul' genotypes from different sources and with other major cultivated varieties, clarifying the number and characteristics of variant sites. The development of a specific fingerprint identification technology using SNP and indel variations in the "Mejhoul" variety holds practical application value. The experimental design is scientifically reasonable, and the experimental results are in line with expectations. This is a well-written manuscript that only requires minor revisions.

Some minor issues that need to be addressed are as follows:

L15-16: The source locations of 3 materials were incorrectly labeled; the correct information should be "three genotypes from USA, Morocco and UAE."

L18: Abbreviations (such as SNP, Indel) are not marked with their full names when they first appear. It is recommended to supplement the full names (Single Nucleotide Polymorphism, Insertion/Deletion) when these abbreviations first occur.

L18-19: The term "cutivar-specific" contains a spelling error and should be corrected to "cultivar-specific".

L22: In the abstract, additional information about the distribution characteristics of variant sites (e.g., whether they are concentrated in coding regions or non-coding regions) can be supplemented; including this information will enhance the depth of the results. The statement "‘Mejhoul’ was derived much earlier" is ambiguous. It is recommended to clarify the basis for judging its relative divergence time or evolutionary status.

L54-55: Please specify the specific parameters of the "suitable climatic conditions" in the Jordan Valley (such as average annual temperature, precipitation, and sunshine duration) to enhance the persuasiveness.

L50: When introducing the 'Mejhoul' date variety, supplement its core characteristics such as fruit quality (including fruit size, sweetness, and taste) and stress resistance, so as to support the statement that it is a "high-quality date variety". The vocabulary “more that 80%” is incorrect and should be revised to “more than 80%”.

L70: "additioanl" (spelling error) should be corrected to "additional".

L74-78: When introducing DNA fingerprinting technology, supplement the types of molecular markers used (such as SSR and SNP), and briefly explain the principle by which they reveal genetic diversity.

L95-96: Please explain the number of genes on the H chain and the L chain respectively.

L97: Revise the label “unknown organism” at the exact center of Figure 2 to "Mejhoul".

L168-170: The base differences and gap differences described in Figure 6 are exactly opposite to the results in the main text. Meanwhile, there is an error in the statistical range of the data. If the caption description in Figure 6 is correct, then the results should be: ‘Berhee’ displayed the most abundant differences with all date palm cultivars (17783-17992) and also gaps (11028-11185). This was followed by ‘Sultana’ with moderate bp differences (194-360) and gaps (119-212) as compared with other date palm cultivars except for the most different cultivar ‘Berhee’.

L188: There is a spelling error in the variety name; "Meghoul" should be corrected to "Mejhoul".

L232: Correct the expression "In the contrary" to "On the contrary". Attempt to analyze the reasons for the differences from previous research results.

L263: Please provide sequencing data of 5 samples (Mejhoul accessions 1-5) in NCBI and describe the basic information of the data.

L268: Figure 7 shows the sequencing coverage of ‘Mejhoul’ accessions 1-5 (It is not just limited to accessions 1-2.).

L302: The supplementary materials are not found throughout the text.

Author Response

After reviewing the manuscript, the evaluation comments are as follows. In this study, next-generation sequencing (NGS) technology was employed to complete the plastid genome assembly of the well-known date palm variety 'Mejhoul', accurately presenting key foundational data such as genome size, GC content, and gene count. Comparative analyses of the plastid genome were conducted between 'Mejhoul' genotypes from different sources and with other major cultivated varieties, clarifying the number and characteristics of variant sites. The development of a specific fingerprint identification technology using SNP and indel variations in the "Mejhoul" variety holds practical application value. The experimental design is scientifically reasonable, and the experimental results are in line with expectations. This is a well-written manuscript that only requires minor revisions.

  • Dear respected reviewer, we highly appreciate your comments and efforts of reviewing our manuscript. We also thank you for the positive comments:

 “The experimental design is scientifically reasonable, and the experimental results are in line with expectations. This is a well-written manuscript that only requires minor revisions.”

Some minor issues that need to be addressed are as follows:

L15-16: The source locations of 3 materials were incorrectly labeled; the correct information should be "three genotypes from USA, Morocco and UAE."

  • Thank you for this comment. The correction was made.

L18: Abbreviations (such as SNP, Indel) are not marked with their full names when they first appear. It is recommended to supplement the full names (Single Nucleotide Polymorphism, Insertion/Deletion) when these abbreviations first occur.

  • Thank you for this important comment, they were written in full the first time they appeared in the Abstract and in text.

L18-19: The term "cutivar-specific" contains a spelling error and should be corrected to "cultivar-specific".

  • Thank you, it was corrected.

L22: In the abstract, additional information about the distribution characteristics of variant sites (e.g., whether they are concentrated in coding regions or non-coding regions) can be supplemented; including this information will enhance the depth of the results. The statement "‘Mejhoul’ was derived much earlier" is ambiguous. It is recommended to clarify the basis for judging its relative divergence time or evolutionary status.

  • Thank you for these two important comments:
  • For the location of variants, yes it is important and was added.
  • For the statement "‘Mejhoul’ was derived much earlier", you are right, it is ambiguous. Therefore it was re-written as follows: The phylogenetic analysis places Mejhoul as a derived lineage within Clade I rather than as an early-diverging cultivar, suggesting it shares a more recent common ancestor with Deglet Noor and Barhee

L54-55: Please specify the specific parameters of the "suitable climatic conditions" in the Jordan Valley (such as average annual temperature, precipitation, and sunshine duration) to enhance the persuasiveness.

  • Thank you for indicating this point, detailed "suitable climatic conditions" were added.

L50: When introducing the 'Mejhoul' date variety, supplement its core characteristics such as fruit quality (including fruit size, sweetness, and taste) and stress resistance, so as to support the statement that it is a "high-quality date variety". The vocabulary “more that 80%” is incorrect and should be revised to “more than 80%”.

  • Important comment, it was added. And “that” was corrected.

L70: "additioanl" (spelling error) should be corrected to "additional".

  • Was corrected

L74-78: When introducing DNA fingerprinting technology, supplement the types of molecular markers used (such as SSR and SNP), and briefly explain the principle by which they reveal genetic diversity.

  • Was added, which would enlighten the reader

L95-96: Please explain the number of genes on the H chain and the L chain respectively.

  • They were corrected for out and inner circles (strands)

L97: Revise the label “unknown organism” at the exact center of Figure 2 to "Mejhoul".

  • Was corrected

L168-170: The base differences and gap differences described in Figure 6 are exactly opposite to the results in the main text. Meanwhile, there is an error in the statistical range of the data. If the caption description in Figure 6 is correct, then the results should be: ‘Berhee’ displayed the most abundant differences with all date palm cultivars (17783-17992) and also gaps (11028-11185). This was followed by ‘Sultana’ with moderate bp differences (194-360) and gaps (119-212) as compared with other date palm cultivars except for the most different cultivar ‘Berhee’.

  • We are deeply thankful for correcting these numbers, they were correct accordingly

L188: There is a spelling error in the variety name; "Meghoul" should be corrected to "Mejhoul".

  • Corrected, thank you

L232: Correct the expression "In the contrary" to "On the contrary". Attempt to analyze the reasons for the differences from previous research results.

  • Corrected, thank you

L263: Please provide sequencing data of 5 samples (Mejhoul accessions 1-5) in NCBI and describe the basic information of the data.

  • Sure, the sequences were under submission and revision under the number: BankIt3015071 and accession number PX560697 (for ‘Mejhoul’ 2)
  • But due the recently ended US governmental shutdown, things are going very slowly at the Genbank

L268: Figure 7 shows the sequencing coverage of ‘Mejhoul’ accessions 1-5 (It is not just limited to accessions 1-2.).

  • Corrected, thank you

L302: The supplementary materials are not found throughout the text

  • Corrected, not available

Reviewer 2 Report

Comments and Suggestions for Authors

The manuscript ‘The complete plastome of 'Mejhoul' date palm provides unique fingerprint for varietal identification’ by Monther T. Sadder, Anfal Alashoush, Nihad Alsmairat, and Anwar Haddad aims to sequence and analyze the complete plastome of the 'Mejhoul' date palm cultivar from different geographic locations. It identifies genetic diversity among 'Mejhoul' genotypes, compares them with major date palm cultivars, and establishes unique SNP and indel markers for cultivar authentication. The study constructs phylogenetic trees to explore evolutionary relationships and discusses genetic diversity implications despite vegetative propagation. Strengths include the use of next-generation sequencing, comprehensive plastome analysis, and potential application in varietal identification. However, several weaknesses must be addressed before publication.

General comments:

The authors conflate traditional DNA fingerprinting with NGS-based genomic profiling. DNA fingerprinting generally uses targeted marker analysis (STRs, RFLP, AFLP), whereas this study uses whole plastome sequencing to identify SNPs and indels, representing a different approach with distinct advantages and limitations. While NGS provides detailed genomic profiles, it is not "classical" DNA fingerprinting. Avoiding the term "fingerprinting" in the context of plastome sequencing would improve conceptual clarity.

Additional validation of identified markers using independent methods or larger, geographically diverse samples is needed. The discussion should critically address how plastome data translates to practical varietal authentication and the pitfalls of interpreting phylogenies based solely on plastid DNA.

Title

The phrase “provides unique fingerprint” is misleading given the use of plastome sequencing rather than classical DNA fingerprinting. "Fingerprint" could be replaced with “genomic markers” or “molecular markers” to better reflect the approach, e.g., “The complete plastome of 'Mejhoul' date palm: genomic markers for varietal identification.”

Introduction

The introduction lacks sufficient justification for the study objective. It focuses heavily on the economic importance of 'Mejhoul' date palm but provides limited information on prior cultivar profiling work or the specific limitations the current study aims to address. The rationale for using costly NGS over classical methods for cultivar identification should be clearer. Currently, the objective (lines 56–58) appears isolated without adequate justification, resembling “research for research's sake.” The introduction should also better differentiate DNA fingerprinting from NGS-based plastome profiling.

Results

A major weakness is the lack of verification of NGS-identified SNPs and indels via gold-standard methods like Sanger sequencing. NGS is known to have limitations in accurately calling such variants, yet no independent validation (e.g., PCR or marker-specific assays) is presented.

The authors misinterpret bootstrap support in phylogenetic analyses. Groups (clades) with low bootstrap values are extensively discussed and even highlighted (Figure 5). Interpretation should be limited to clades with bootstrap support ≥70. Incorporating additional statistical support, like posterior probabilities, would strengthen conclusions. Also, the maximum likelihood method used does not account for indel distributions, which is critical given the study’s focus.

Discussion

Methodologically, the authors should clarify that their approach does not replace fingerprinting but rather complements it with high-resolution genomic data. The lack of discussion on NGS limitations—such as sequencing errors, coverage biases, and bioinformatics complexities—weaken the technical rigor. Moreover, focusing solely on plastid genomes overlooks nuclear genome variation, potentially missing important diversity signals in a dioecious species like date palm.

It is also unclear why the authors consider only female plants as the limiting factor for determining varietal characteristics, ignoring paternal inheritance, which contributes nearly half of the genetic material (lines 209–210).

Despite the extensive discussion, the conclusion regarding the causes of observed plastome heterogeneity remains unclear and seems disconnected from the main text (lines 206–225).

Furthermore, making evolutionary inferences (including relationships) from unrooted trees is difficult. At minimum, an outgroup should be included—particularly in Figures 3 and 4, which are based on limited samples. Given the low support values, absence of outgroups, poor resolution of some branches, and the maximum likelihood method’s inability to account for indels, drawing firm conclusions from these figures is virtually impossible.

Materials and Methods

Methods omit details on sequencing depth thresholds and error correction steps. Expand on bioinformatics tools and parameters beyond software names for sequence alignment and tree construction.

Conclusion

The conclusion is overly brief and should better summarize findings and implications.

Specific Comments:

  • Use either “indels” or “InDels” consistently; “indels” is more standard. 
  • Lines 27-28: Clarify “one of 14 species”—14 species of what group?
  • Line 90: Replace “The major object of this study” with “The major objective of this study.”
  • Figure 1. The figure’s relevance and intended conclusions are unclear.
  • Figure 2. Please specify what is meant by the label “unknown organism” on the figure.
  • Table 1: Add color legend; remove yellow highlighting in the header.
  • Table 2: Improve header formatting; remove yellow highlighting.

Author Response

The manuscript ‘The complete plastome of 'Mejhoul' date palm provides unique fingerprint for varietal identification’ by Monther T. Sadder, Anfal Alashoush, Nihad Alsmairat, and Anwar Haddad aims to sequence and analyze the complete plastome of the 'Mejhoul' date palm cultivar from different geographic locations. It identifies genetic diversity among 'Mejhoul' genotypes, compares them with major date palm cultivars, and establishes unique SNP and indel markers for cultivar authentication. The study constructs phylogenetic trees to explore evolutionary relationships and discusses genetic diversity implications despite vegetative propagation. Strengths include the use of next-generation sequencing, comprehensive plastome analysis, and potential application in varietal identification. However, several weaknesses must be addressed before publication.

  • Dear respected reviewer, we highly appreciate your comments and efforts of reviewing our manuscript. We also thank you for the positive comments:

 “Strengths include the use of next-generation sequencing, comprehensive plastome analysis, and potential application in varietal identification.”

General comments:

The authors conflate traditional DNA fingerprinting with NGS-based genomic profiling. DNA fingerprinting generally uses targeted marker analysis (STRs, RFLP, AFLP), whereas this study uses whole plastome sequencing to identify SNPs and indels, representing a different approach with distinct advantages and limitations. While NGS provides detailed genomic profiles, it is not "classical" DNA fingerprinting. Avoiding the term "fingerprinting" in the context of plastome sequencing would improve conceptual clarity.

  • Thank you for this comment, we clarified using finger printing in text by adding most used DNA marker SSR.

Additional validation of identified markers using independent methods or larger, geographically diverse samples is needed. The discussion should critically address how plastome data translates to practical varietal authentication and the pitfalls of interpreting phylogenies based solely on plastid DNA.

  • Thank you very much for this comment. In fact, for plant plastomes, the norm is to use just one sample per cultivar and this is why we would criticize them, as we have more plastome samples for one cultivar, in our study, we have eight Mejhoul samples, which is the new in any plant plastome study, some recent studies (with five samples in olives for example) found similar variations). Therefore, this would be a major recommendation for NGS studies using plastome analysis.

Title

The phrase “provides unique fingerprint” is misleading given the use of plastome sequencing rather than classical DNA fingerprinting. "Fingerprint" could be replaced with “genomic markers” or “molecular markers” to better reflect the approach, e.g., “The complete plastome of 'Mejhoul' date palm: genomic markers for varietal identification.”

  • Highly appreciate your suggestion, which was used in the revised title.

Introduction

The introduction lacks sufficient justification for the study objective. It focuses heavily on the economic importance of 'Mejhoul' date palm but provides limited information on prior cultivar profiling work or the specific limitations the current study aims to address. The rationale for using costly NGS over classical methods for cultivar identification should be clearer. Currently, the objective (lines 56–58) appears isolated without adequate justification, resembling “research for research's sake.” The introduction should also better differentiate DNA fingerprinting from NGS-based plastome profiling.

  • Thank you for this crucial comment. Two paragraphs were added to the introduction to clarify this issue

Results

A major weakness is the lack of verification of NGS-identified SNPs and indels via gold-standard methods like Sanger sequencing. NGS is known to have limitations in accurately calling such variants, yet no independent validation (e.g., PCR or marker-specific assays) is presented.

  • Thank you for this comment. This was not an issue as we have a huge sequencing depth (Figure 7) upto 1000-2000x. and this is the norm for plastome studies , the depth of thousands is more reliable than Sanger sequencing for one or two clones.

The authors misinterpret bootstrap support in phylogenetic analyses. Groups (clades) with low bootstrap values are extensively discussed and even highlighted (Figure 5). Interpretation should be limited to clades with bootstrap support ≥70. Incorporating additional statistical support, like posterior probabilities, would strengthen conclusions. Also, the maximum likelihood method used does not account for indel distributions, which is critical given the study’s focus.

  • Very important comment. We have included an outgroup to better resolve the maximum likelihood tree and revised the results.

Discussion

Methodologically, the authors should clarify that their approach does not replace fingerprinting but rather complements it with high-resolution genomic data. The lack of discussion on NGS limitations—such as sequencing errors, coverage biases, and bioinformatics complexities—weaken the technical rigor. Moreover, focusing solely on plastid genomes overlooks nuclear genome variation, potentially missing important diversity signals in a dioecious species like date palm.

  • Thank you for your comment. In contrast, the plastome (chloroplast genome) is highly suitable for differentiating female lineages due to its uniparental (maternal) inheritance, conserved structure, and moderate evolutionary rate. Plastid genomes provide a stable framework for assessing variation among cultivars that share maternal genomic backgrounds. With the advent of next-generation sequencing (NGS), it is now possible to obtain complete plastome sequences with high depth and accuracy. Compared with classical markers, NGS-based plastome profiling provides maternally inherited variations, enabling the development of precise and robust diagnostic markers for cultivar authentication, especially for elite female clones like ‘Mejhoul’.

It is also unclear why the authors consider only female plants as the limiting factor for determining varietal characteristics, ignoring paternal inheritance, which contributes nearly half of the genetic material (lines 209–210).

  • That is correct, because the generated fruits are ONLY the makeup of the female genome.

Despite the extensive discussion, the conclusion regarding the causes of observed plastome heterogeneity remains unclear and seems disconnected from the main text (lines 206–225).

  • Thank you, a major conclusion paragraph was added.

Furthermore, making evolutionary inferences (including relationships) from unrooted trees is difficult. At minimum, an outgroup should be included—particularly in Figures 3 and 4, which are based on limited samples. Given the low support values, absence of outgroups, poor resolution of some branches, and the maximum likelihood method’s inability to account for indels, drawing firm conclusions from these figures is virtually impossible.

  • Very important comment. We have included an outgroup to better resolve the maximum likelihood tree.

Materials and Methods

Methods omit details on sequencing depth thresholds and error correction steps. Expand on bioinformatics tools and parameters beyond software names for sequence alignment and tree construction.

  • Were added, thank you

Conclusion

The conclusion is overly brief and should better summarize findings and implications.

  • Thank you, a major conclusion paragraph was added.

Specific Comments:

  • Use either “indels” or “InDels” consistently; “indels” is more standard. 
  •  was unified
  • Lines 27-28: Clarify “one of 14 species”—14 species of what group?
  •  of the genus Phoenix, was added
  • Line 90: Replace “The major object of this study” with “The major objective of this study.”
  •  was corrected
  • Figure 1. The figure’s relevance and intended conclusions are unclear.
  •  Thank you, its important to show the tree structure and fruit shape for investigated species for clarity.
  • Figure 2. Please specify what is meant by the label “unknown organism” on the figure.
  •  was corrected, thank you
  • Table 1: Add color legend; remove yellow highlighting in the header.
  • yellow highlighting was removed
  • Table 2: Improve header formatting; remove yellow highlighting.
  • yellow highlighting was removed

Round 2

Reviewer 2 Report

Comments and Suggestions for Authors

I have reviewed the revised version of the manuscript and appreciate the authors’ responses and efforts in addressing the corrections. However, I am not fully satisfied with the answers to all questions; some issues still require further revision. The most critical points are as follows:

1. The description of the bioinformatic methods remains insufficient. Key parameters for alignment and tree construction are still missing, making it impossible to replicate the authors’ approach based on the current description.

2. The presentation and interpretation of the phylogenetic trees are particularly problematic. Incorrect interpretation of nodes and support values often leads to speculative conclusions. The authors continue to discuss statistically unsupported nodes, which is unacceptable. I recommend the authors devote more time to understanding the correct principles of phylogenetic tree interpretation. Additionally, some trees include too few samples for objective analysis. Our previous comment regarding the addition of an outgroups, especially in cases of small sample sizes, has not been addressed. Specifically:

- Figure 3 and its description in the Results section: The lack of an outgroup prevents accurate inference of the evolutionary relationships in the group. Low bootstrap values of some clades and unresolved branches (lack of fully bifurcating nodes) lead to speculative interpretations.

In particular:

Line 144: The authors’ claim that “‘Mejhoul’ 1 accession was the first to be resolved in the tree” is incorrect due to the absence of an outgroup and lack of bootstrap support.

Line 144: The statement “followed by accession 4, then 3, and followed by 5 and 2” is not supported, as relationships among these accessions remain unresolved with low support values and lack of strict bifurcation division, especially lines 3 and 4.

Line 147: The conclusions rely on low bootstrap support, weakening their validity.

The cluster containing accessions 6, 7, and 8, described as a strongly supported group (100% bootstrap), is noted; however, the relationship between ‘Mejhoul’ 7 and 8 as a subclade, and ‘Mejhoul’ 6 grouped with a male genotype appearing later, needs clearer support.

– Figure 4 and its description exhibit similar issues to Figure 3:

Lines 175–176: The assertion that “the resolved big cluster showed that ‘Khalas’ was the first to emerge in the cluster, followed by ‘DP00001’ from Pakistan” is speculative, given the lack of bootstrap support and absence of an outgroup.

– Figure 5 and its interpretation are especially concerning:

In particular:

Group 1 has very low statistical support.

Groups 2 and 3 are not clades at all yet are treated as such in the discussion.

Therefore, the entire section covering lines 184–195 and the corresponding discussion should be thoroughly re-evaluated.

3. The conclusion has been expanded per my earlier comments but remains insufficient. It should succinctly summarize the main findings rather than presenting lengthy arguments.

4. Table 1 still lacks the requested color legend.

Author Response

I have reviewed the revised version of the manuscript and appreciate the authors’ responses and efforts in addressing the corrections. However, I am not fully satisfied with the answers to all questions; some issues still require further revision. The most critical points are as follows:

--> We highly appreciate the respected reviewer’s time and efforts for the constructive comments.

  1. The description of the bioinformatic methods remains insufficient. Key parameters for alignment and tree construction are still missing, making it impossible to replicate the authors’ approach based on the current description.

--> thank you for this important issue, all alignment and tree construction parameters were included.

  1. The presentation and interpretation of the phylogenetic trees are particularly problematic. Incorrect interpretation of nodes and support values often leads to speculative conclusions. The authors continue to discuss statistically unsupported nodes, which is unacceptable. I recommend the authors devote more time to understanding the correct principles of phylogenetic tree interpretation. Additionally, some trees include too few samples for objective analysis. Our previous comment regarding the addition of an outgroups, especially in cases of small sample sizes, has not been addressed. Specifically:

--> thank you for this comment, outgroups were added to all trees

- Figure 3 and its description in the Results section: The lack of an outgroup prevents accurate inference of the evolutionary relationships in the group. Low bootstrap values of some clades and unresolved branches (lack of fully bifurcating nodes) lead to speculative interpretations.

--> an outgroup was added and related results revised

In particular:

Line 144: The authors’ claim that “‘Mejhoul’ 1 accession was the first to be resolved in the tree” is incorrect due to the absence of an outgroup and lack of bootstrap support.

--> Corrected as advised

Line 144: The statement “followed by accession 4, then 3, and followed by 5 and 2” is not supported, as relationships among these accessions remain unresolved with low support values and lack of strict bifurcation division, especially lines 3 and 4.

--> Corrected as advised

Line 147: The conclusions rely on low bootstrap support, weakening their validity.

--> Corrected as advised

The cluster containing accessions 6, 7, and 8, described as a strongly supported group (100% bootstrap), is noted; however, the relationship between ‘Mejhoul’ 7 and 8 as a subclade, and ‘Mejhoul’ 6 grouped with a male genotype appearing later, needs clearer support.

--> Corrected as advised

– Figure 4 and its description exhibit similar issues to Figure 3:

Lines 175–176: The assertion that “the resolved big cluster showed that ‘Khalas’ was the first to emerge in the cluster, followed by ‘DP00001’ from Pakistan” is speculative, given the lack of bootstrap support and absence of an outgroup.

--> Corrected as advised

– Figure 5 and its interpretation are especially concerning:

--> Corrected as advised

In particular:

Group 1 has very low statistical support.

Groups 2 and 3 are not clades at all yet are treated as such in the discussion.

Therefore, the entire section covering lines 184–195 and the corresponding discussion should be thoroughly re-evaluated.

--> Corrected as advised

  1. The conclusion has been expanded per my earlier comments but remains insufficient. It should succinctly summarize the main findings rather than presenting lengthy arguments.

--> thank you, it was revised as requested.

  1. Table 1 still lacks the requested color legend.

--> It was added, thank you

Round 3

Reviewer 2 Report

Comments and Suggestions for Authors

I have reviewed the revised manuscript and appreciate the authors’ responses and efforts to address the corrections. Indeed, some of my previous comments have been adequately addressed. However, the presentation and interpretation of the phylogenetic trees, particularly Figure 5, remain problematic. Misinterpretation of the indicated groups and support values often leads to speculative conclusions. The authors continue to discuss statistically unsupported nodes and irrelevant groups, which is unacceptable.

The distinction of three groups in this study is fundamentally incorrect. Group 1 lacks statistical support (bootstrap value is 46) and therefore cannot be treated as a valid group. Groups 2 and 3 are not clades at all and must not be referred to as such. These are composite groups that do not share a common ancestor (common node) and even include outgroups! This interpretation contradicts the fundamental principle of obligatory monophyly in phylogenetic analysis and is a serious error. Both Figure 5 and its description (lines 195–205) require correction.

The manuscript should only be considered for publication once these misinterpretations, which risk misleading readers, are fully corrected.